

# Exploring the anti-aging effects of fisetin in telomerase-deficient progeria mouse model

Rui Zhao[1,*], Haomeng Kou[1,*], Duo Jiang[1] and Feng Wang[1,2]

[1] Department of Genetics, Tianjin Medical University, Tianjin, China
[2] Institute of Prosthodontics School and Hospital of Stomatology, Tianjin Medical University, Tianjin, China
[*] These authors contributed equally to this work.

## ABSTRACT

Aging is a natural and complex process characterized by the gradual deterioration of tissue and physiological functions in the organism over time. Cell senescence, a hallmark of aging, refers to the permanent and irreversible cell cycle arrest of proliferating cells triggered by endogenous stimuli or environmental stresses. Eliminating senescent cells has been shown to extend the healthy lifespan. In this study, we established a progeria mouse model with telomerase deficiency and confirmed the presence of shortened telomere length and increased expression of aging markers $p16^{INK4a}$ and $p21^{CIP1}$ in the organ tissues of G3 $Tert^{-/-}$ mice. We identified fisetin as a potent senolytic drug capable of reversing premature aging signs in telomerase-deficient mice. Fisetin treatment effectively suppressed the upregulation of aging markers $p16^{INK4a}$ and $p21^{CIP1}$ and reduced collagen fiber deposition. Furthermore, we observed a significant elevation in the mRNA level of $Stc1$ in G3 $Tert^{-/-}$ mice, which was reduced after fisetin treatment. Stc1 has been implicated in anti-apoptotic processes through the upregulation of the Akt signaling pathway. Our findings reveal that fisetin exerts its anti-aging effect by inhibiting the Akt signaling pathway through the suppression of $Stc1$ expression, leading to the apoptosis of senescent cells.

## INTRODUCTION

Aging is a complex and progressive process characterized by the gradual deterioration of tissue integrity and physiological functions in the body (*Lopez-Otin et al., 2023*). This natural phenomenon ultimately leads to the onset of age-related diseases and, eventually, death (*Leidal, Levine & Debnath, 2018*). While advancements in public health and medical technology have significantly increased the average life expectancy globally, the aging population continues to grow at an unprecedented rate. According to the United Nations's projections, the number of individuals aged 60 and above is expected to reach 2.1 billion by 2050 (*Sun, Li & Kirkland, 2022*). The implications of aging, as a major risk factor for various chronic diseases (*Goldberg & Dixit, 2015*), highlight the urgent need for innovative medical interventions and healthcare models. Failing to address this issue effectively will impose an irreversible burden of chronic diseases, with significant societal and economic

Corresponding author
Feng Wang, wangf@tmu.edu.cn

consequences (*Scott, Ellison & Sinclair, 2021*). Therefore, it is of utmost importance to delve into the mechanisms of aging and develop strategies that promote healthy aging.

Recent studies have revealed 12 common features of aging in organisms, including cell senescence, telomere attrition, chronic inflammation, and more (*Lopez-Otin et al., 2023*). Telomeres, often referred to as the "aging clock", are located at the ends of eukaryotic chromosomes and consist of repetitive nucleotide sequences (TTAGGG), along with the telomere shelterin complex (*Giardini et al., 2014*). Serving as protective caps for chromosome ends, telomeres play a critical role in maintaining the structural integrity of chromosomes and genome stability (*Turner, Vasu & Griffin, 2019*). With each round of cell division, telomeres naturally undergo gradual shortening due to the end replication problem (*Harley, Futcher & Greider, 1990*; *Olovnikov, 1973*; *Watson, 1972*). Telomerase, comprising the RNA component (Terc) and reverse transcriptase (Tert), is responsible for preserving telomere length (*Shay & Wright, 2019*; *Wu et al., 2017*). To investigate the correlation between telomere dysfunction and age-related diseases, researchers have utilized telomerase-deficient mice as models.

Research has demonstrated that targeted clearance of senescent cells can extend lifespan and has shed light on the significant role of cellular senescence in driving aging and the development of age-related diseases (*Baker et al., 2011*; *Demaria et al., 2014*). In 2015, the first senolytic drugs were reported, and since then, an increasing number of drugs have been utilized in aging models. One notable example is the combination of dasatinib and quercetin (D+Q), which has shown effective clearance of senescent cells (*Xu et al., 2018*). Additionally, the small molecule inhibitor ABT-263, which acts on anti-apoptotic proteins, has been found to induce lysis in senescent human lung fibroblasts (*Yosef et al., 2016*). Fisetin, a polyphenolic compound naturally present in fruits and vegetables (*Khan et al., 2013*), was discovered and has garnered attention as an effective senolytic drug.

Emerging evidence suggests that fisetin exhibits significant senolytic properties. In senescent mouse embryonic fibroblasts, fisetin reduces the number of senescent cells in a dose-dependent manner, surpassing the therapeutic effects of quercetin. Intermittent administration of fisetin through the diet has been shown to sustainably suppress the expression of aging markers in various organ tissues, alleviate age-related tissue pathologies, and ultimately extend median lifespan and longevity in mice (*Yousefzadeh et al., 2018*). However, the mechanism behind fisetin's anti-aging effects in premature aging mouse models remains unclear and warrants further exploration.

Our study aimed to establish a premature aging mouse model with *Tert* knockout (G3 $Tert^{-/-}$) and investigate the effects of fisetin treatment. We confirmed that the telomere length was significantly shortened, and the expression of senescence markers $p16^{INK4a}$ and $p21^{CIP1}$ was significantly increased in the kidney and liver tissues of G3 $Tert^{-/-}$ mice. To assess the effectiveness of fisetin, we administered it through intermittent gavage and continuous dietary treatment. Blood biochemistry analysis revealed that fisetin treatment effectively inhibited alanine transaminase (ALT) and urea levels. Furthermore, fisetin significantly reduced the expression of senescence markers in the kidney and liver tissues of G3 $Tert^{-/-}$ mice. Additionally, Masson staining results demonstrated that fisetin treatment effectively reduced collagen fiber deposition in the kidney tissues of G3

$Tert^{-/-}$ mice. RNA sequencing and RT-PCR analysis showed a significant increase in the expression of $Stc1$ in G3 $Tert^{-/-}$ mice, which was decreased after fisetin treatment. Stc1 is highly expressed in various tumor tissues and regulates the invasion and metastasis of cancer cells by modulating the PI3K/Akt signaling pathway. Our results also revealed that G3 $Tert^{-/-}$ mice exhibited a significant upregulation of pAkt and its downstream anti-apoptotic protein Bcl-2 compared to wild-type (WT) mice. However, these levels decreased to lower levels following fisetin treatment. These findings suggest that fisetin exerts its senolytic effects by suppressing the expression of $Stc1$ and inhibiting the activation of the Akt signaling pathway in G3 $Tert^{-/-}$ mice.

## MATERIALS AND METHODS

### Animals and fisetin treatment

The WT male C57BL/6J mice were purchased from the Academy of Military Science (Beijing, China). $Tert^{+/-}$ mice were a gift from Professor Yusheng Cong and were crossed to obtain $Tert^{-/-}$ mice. All mice were maintained under specific pathogen-free conditions and fed a full-price diet and provided with autoclaved water. Each cage was housing 3 mice. At 10 weeks of age, G3 $Tert^{-/-}$ mice were divided into vehicle group and fisetin treatment group using a random number table. G3 $Tert^{-/-}$ + Fisetin group received a diet supplemented with 500 mg/kg of fisetin, and they were also subjected to gastric gavage with 20 mg/kg of fisetin 3 times a week for a total of 7 weeks. The mice were fasted for 6–8 h before conducting the gavage procedure. The WT and remaining G3 $Tert^{-/-}$ groups received a placebo treatment. After a 12 h fasting period, the mice were anesthetized using isoflurane *via* inhalation, blood was collected from the orbital sinus, allowed to stand at room temperature for 2 h, and then centrifuged at $1000 \times g$ for 30 min to obtain serum, which was sent for examination to analyze liver and kidney function. Then mice were euthanized using $CO_2$, and their liver and kidneys were harvested for further analysis. Only male mice were used in this study. Fisetin was purchased from meilunbio (Dalian, China). According to the instructions, dissolve fisetin in 10% ethanol, 30% PEG 400, and 60% Phosal 50 pg for oral gavage treatment. All experiments were approved by the Animal Ethics Committee of Tianjin Medical University (TMUaMEC2022015).

### Southern blot

The DNA was extracted from the kidney tissues of WT, G1 $Tert^{-/-}$, and G3 $Tert^{-/-}$ mouse, and its concentration was measured. Then, 3 μg of DNA were loaded onto a 1% agarose gel for pulse-field gel electrophoresis under the following conditions: 14 °C, 16 h, ts1 =1s, ts2 =2s, U = 6v/cm. After electrophoresis, the gel was transferred to nylon membrane using negative pressure filtration. The gel was then cross-linked twice using UV light at $1,200 \times 100$ μJ/cm$^2$ energy. The hybridization solution was incubated at 42 °C overnight with 1:500 DIG-labeled probes during the first use, and 1:1000 probe ratio in subsequent uses. The nylon membrane was washed vigorously twice with 2×SSC and 0. 2×SSC containing 0.1% SDS at 42 °C, and then washed once with 1×washing buffer at room temperature. Afterward, the membrane was incubated with DIG antibodies for 5 h,

followed by one wash with $1\times$ washing buffer and $1\times$ detection buffer before visualization and photography.

## RNA extraction and sequencing (RNA-Seq)

Total RNA was extracted from different groups of mice kidney or liver tissue ($n = 3$) using Trizol. Each sample represents one kidney tissue isolated from a single mouse, and each group consists of three samples. RNA samples with an RNA integrity value (RIN) $\geq 7.0$ and 28S/18S ratio $\geq 1.5$ were selected for sequencing. The experimental design involved statistical power analysis using the RNASeqPower software. After quality control and filtering of the sequencing data, the filtered clean reads were aligned to the reference sequences to determine the alignment pass rate in the second round of quality control (QC of alignment) and to perform differential gene expression analysis. A total of 9 cDNA libraries were constructed and the BGISEQ-500 platform was used for RNA sequencing. Differential gene expression analysis was performed using the DESeq2 software. The *P*-values were then adjusted for multiple testing using the false discovery rate (FDR) method, with a threshold of *Q*-value $\leq 0.05$ considered statistically significant. Heatmap analysis was performed using Bowtie2 to compare clean reads with the reference gene sequences, and RSEM was used to calculate the gene expression levels for each sample. The DESeq2 method is based on the principles of negative binomial distribution.

## Quantitative real-time PCR (RT-PCR)

According to the manufacturer's protocol, 12 RNA samples from the WT + Vehicle, WT + Fisetin, G3 $Tert^{-/-}$ + Vehicle and G3 $Tert^{-/-}$ + Fisetin groups were reverse transcribed into cDNA using ABScript III Reverse Transcriptase (ABclonal, Wuhan, China). RT-PCR analysis was performed using the $2\times$ Universal SYBR Green Fast qPCR Mix (ABclonal, Wuhan, China) and mouse *glyceraldehyde-3-phosphate dehydrogenase* (*Gapdh*) was used for normalization. All primers were synthesized by Sangon Biotech (Shanghai, China) Co., Ltd, and the sequence were listed in Table 1.

## Masson staining

Masson staining was performed following the manufacturer's kit instructions (Solarbio, China). Briefly, after deparaffinization of paraffin-embedded sections with xylene, the sections were stained with a prepared Weigert's iron hematoxylin stain solution (Weigert A: Weigert B = 1:1) for 10 min. After differentiation in acidic ethanol differentiation solution for 10 s, they were rinsed with water. The sections were treated with Masson's blue staining solution for 5 min, followed by a water rinse. Distilled water was used for a 1 min wash. After staining with Aniline Blue stain solution for 10 min, a weak acid working solution was used for a 1 min wash. The sections were washed with phosphomolybdic acid for 2 min and then with the weak acid working solution for 1 min again. After staining in Aniline Blue staining solution for 1 min, the sections were washed again with the weak acid working solution for 1 min. They were dehydrated with 95% and absolute ethanol, cleared with xylene three times for 1 min each, and finally mounted with neutral resin. Microscopic examination was carried out to observe and capture representative images of the stained sections.

**Table 1  Sequences of primers used for RT-PCR.**

| Gene | Forward (5′–3′) | Reverse (5′–3′) |
| --- | --- | --- |
| $p16^{INK4a}$ | GAACTCTTTCGGTCGTACCC | AGTTCGAATCTGCACCGTAGT |
| $p21^{CIP1}$ | GTCAGGCTGGTCTGCCTCCG | CGGTCCCGTGGACAGTGAGCAG |
| Stc1 | AGGAGGACTGCTACAGCAAGCT | TCCAGAAGGCTTCGGACAAGTC |
| Gapdh | AAGGTCATCCCAGAGCTGAA | CTGCTTCACCACCTTCTTGA |

## Western blot

Mice kidney tissues were collected and homogenized in liquid nitrogen. RIPA lysis buffer (Solarbio, Beijing, China) and proteinase inhibitor, PMSF (MedChemExpress, Monmouth Junction, NJ, USA), were added to the tissue lysate, which was then subjected to 30 min of lysis on ice. The lysate was centrifuged at 12,000× g for 20 min at 4 °C, and the supernatant was collected. 5 × SDS loading buffer was added to the supernatant, followed by denaturation at 100 °C for 10 min. The denatured proteins were separated by 10% SDS-PAGE electrophoresis and transferred onto PVDF membrane. The membrane was incubated with the primary antibody overnight at 4 °C, followed by three washes with TBST (25 mM Tris, 150 mM NaCl, 2.7 mM KCl, 0.1% Tween-20) for 10 min each. The membrane was then incubated with the secondary antibody at room temperature for 1 h. Finally, immunoreactivity was detected by chemiluminescence.

## Antibodies

Antibodies used in this study were listed as follows: Akt (9272S, 1:1000; Cell Signaling Technology, Danvers, MA, USA), pAkt Ser473 (4060T, 1:1000; Cell Signaling Technology, Danvers, MA, USA), Bcl-2 (AB112, 1:1000), $\beta$-tubulin (AC008, 1:2000; ABclonal, Woburn, MA, USA).

## Statistical analysis

IBM SPSS Statistics 21 software were used for statistical analysis. The results were expressed as mean ± standard error of mean (SEM). Statistical values ($P$) were obtained using unpaired $t$-test (the data conformed to the normal distribution). $P < 0.05$ was considered statistically significant.

# RESULTS

## Construction of telomerase deficient progeria mouse model

We established G1 $Tert^{-/-}$, and G3 $Tert^{-/-}$ mice to create a telomerase-deficient progeria mouse model (Figs. 1A and 1B). As mice age, the rate of weight gain in G3 $Tert^{-/-}$ mice diminishes compared to that of WT mice (Fig. S1A). Additionally, the fur of G4 $Tert^{-/-}$ mice exhibited a graying of color, coarseness, loss of luster, and gradual thinning, sometimes accompanied by hair loss (Fig. S1B). Telomerase deficiency in $Tert^{-/-}$ mice led to the accumulation of telomere dysfunction. To verify changes in telomere length in $Tert^{-/-}$ mice, Southern blot analysis was conducted. By continuously breeding $Tert^{-/-}$ heterozygous mice for three generations, we observed that the telomere length of G3 $Tert^{-/-}$ mice was shorter than that of WT and G1 $Tert^{-/-}$ mice (Fig. 1C), which is

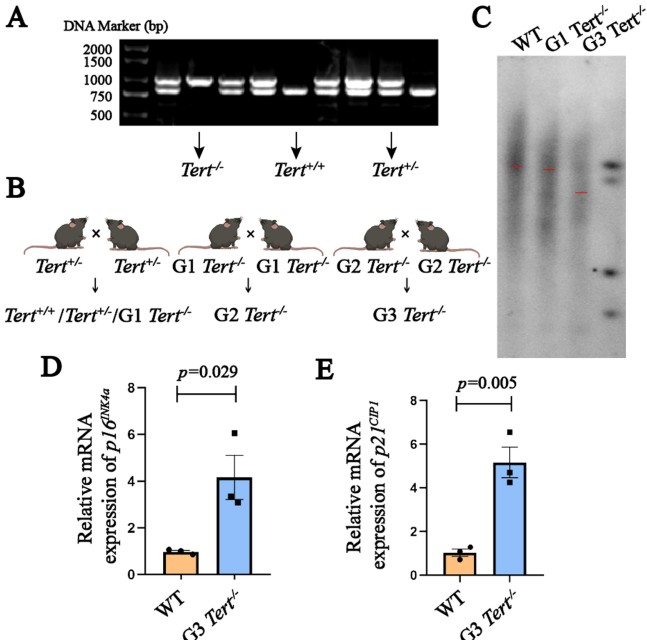

**Figure 1 Construction of telomerase deficient progeria mouse model.** (A) Genotyping of $Tert^{+/-}$ offspring mice was performed using agarose gel electrophoresis. The DNA amplification product of $Tert^{-/-}$ mice appeared as a single band at 1,000 bp, $Tert^{+/+}$ mice showed a band at 800 bp, and $Tert^{+/-}$ mice exhibited two bands at 800 and 1,000 bp. (B) The technical workflow for obtaining G3 $Tert^{-/-}$ premature aging mice. (C) Southern blot analysis to determine telomere length in WT, G1, and G3 $Tert^{-/-}$ mice. (D–E) RT-PCR analysis to measure the mRNA expression levels of senescence markers $p16^{INK4a}$ and $p21^{CIP1}$ in kidney tissue ($n = 3$). Data represents the mean ± SEM using unpaired $t$-test.

consistent with previous findings (*Liu et al., 2000*; *Strong et al., 2011*). Previous studies have shown that $Tert^{-/-}$ mice exhibit a degenerative phenotype resembling premature aging (*Jaskelioff et al., 2011*; *Rudolph et al., 1999*). In our study, we observed high expression levels of aging-associated genes, $p16^{INK4a}$ and $p21^{CIP1}$, in kidney tissues (Figs. 1D and 1E). It indicates that homozygous depletion of *Tert* accelerates telomere shortening and aging in the third generation of knockout mice when compared to WT mice. However, our data revealed that, compared to age-matched WT mice, there were no significant changes in the spleen and thymus indices of G3 $Tert^{-/-}$ mice (Figs. S1C and S1D). This may be attributed to the fact that thymus and spleen indices depend on the extent of lymphocyte proliferation within these organs, making them relatively crude and lagging indicators. Meanwhile, we also observed that there were no significant changes in aging markers and senescence-associated secretory phenotypes (SASP) in the lungs and spleens of G3 $Tert^{-/-}$ mice (Figs. S1E and S1F). This may suggest that aging is a progressive process of declining organismal adaptability and reduced resistance, and it may not necessarily result in significant organic changes in the absence of external stimuli or pathogen exposure.

## Treatment with fisetin improves premature aging symptoms in $Tert^{-/-}$ mice

Accumulating evidence supports the potential of fisetin as a senolytic drug (*Zhu et al., 2017*). To evaluate the anti-aging effects of fisetin on *Tert*-deficient progeria mice, we randomly divided WT and G3 $Tert^{-/-}$ mice into vehicle and fisetin-treated groups, administering fisetin orally continuously and intermittently *via* gavage (Fig. 2A). To assess the impact of fisetin treatment on the liver and kidney, we initially examined the levels of alanine transferase (ALT) and urea through blood biochemistry. The results demonstrated that fisetin treatment effectively mitigated the elevation of serum ALT and urea levels in G3 $Tert^{-/-}$ mice (Figs. 2B and 2C). Additionally, fisetin treatment led to a significant reduction in the mRNA levels of $p16^{INK4a}$ and $p21^{CIP1}$ in the kidney and liver tissues of G3 $Tert^{-/-}$ mice (Figs. 2D–2G). Masson staining revealed a notable increase in collagen fiber deposition in the kidney tissue of G3 $Tert^{-/-}$ mice, which was ameliorated after fisetin treatment (Fig. 2H). These findings collectively indicate that fisetin exhibits potent anti-aging effects in the telomerase-deficient progeria mouse model.

## Screening of potential targets for anti-aging mechanism of fisetin

To gain insights into the potential mechanism underlying the anti-aging effects of fisetin in telomerase-deficient progeria mice, we conducted RNA sequencing on kidney tissues obtained from three groups: WT + Vehicle, G3 $Tert^{-/-}$ + Vehicle, and G3 $Tert^{-/-}$ + Fisetin. The statistical power of this experimental design, calculated in RNASeqPower is 0.99997 (depth = 100, cv = 0.1, effect = 2, $\alpha$ = 0.05, power = 0.99997). Comparative analysis revealed 1735 differentially expressed genes in the G3 $Tert^{-/-}$ + Vehicle group compared to the WT + Vehicle group, with 482 genes upregulated and 1253 genes downregulated (Fig. 3A). Moreover, we identified 388 differentially expressed genes in the G3 $Tert^{-/-}$ + Vehicle group compared to the G3 $Tert^{-/-}$ + Fisetin group, with 251 genes upregulated and 137 genes downregulated (Fig. 3A). The Venn diagram showed an intersection of differential genes, totaling 119 genes, of which 83 genes are upregulated (Fig. 3A), and the expression patterns of the top 20 genes were visualized in the heat map (Fig. 3B).

Gene Ontology (GO) biology process analysis reveals that 83 up-regulated differential genes are primarily enriched in signaling pathways such as blood circulation, production of molecular mediator of immune response, and circulatory system processes (Fig. S2A). Additionally, we conducted a Kyoto Encyclopedia of Genes and Genomes (KEGG) analysis on the differential expression genes in the G3 $Tert^{-/-}$ + Vehicle group compared to the G3 $Tert^{-/-}$ + Fisetin group (Figs. 3C and 3D), revealing intriguing insights. Specifically, the upregulated differential genes in the G3 $Tert^{-/-}$ group exhibited significant enrichment in pathways associated with oxidative phosphorylation, reactive oxygen species (ROS) dynamics, and age-related diseases (Fig. 3C). It is worth noting that aging is characterized by a multitude of factors, including genomic instability, telomere attrition, epigenetic modifications, mitochondrial dysfunction, and more (*Lopez-Otin et al., 2023*). Mitochondria, being pivotal for overall health maintenance, play a central role in this context. With the progression of age, mitochondrial function often declines, leading to heightened ROS production. This excessive ROS generation triggers oxidative

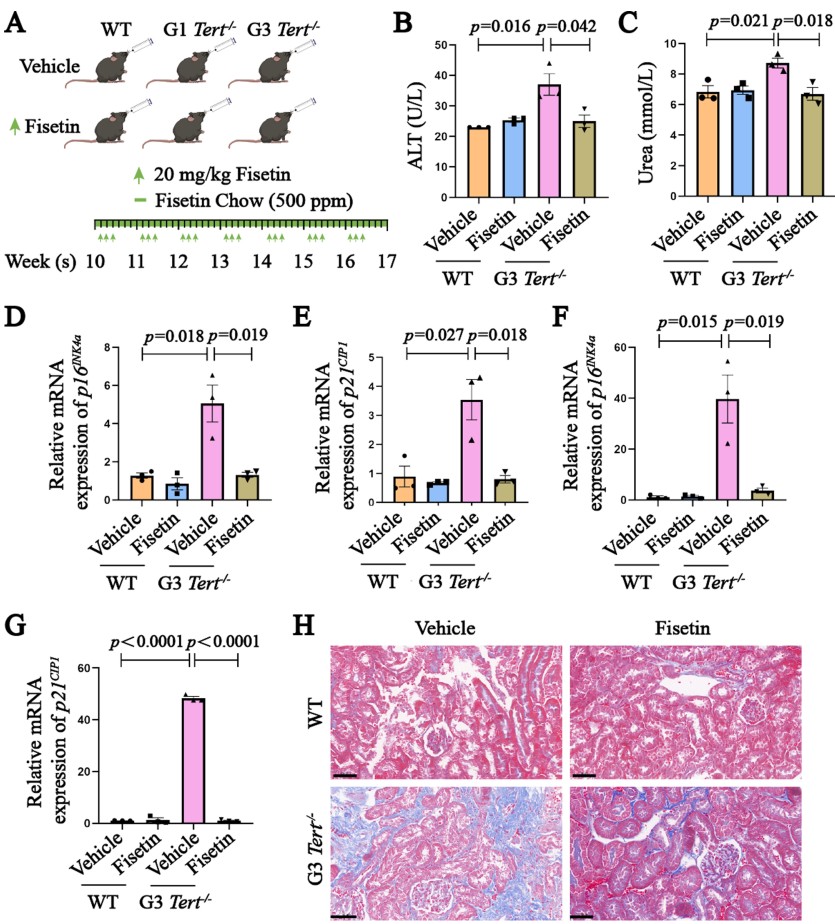

**Figure 2** **Treatment with fisetin improves premature aging symptoms in *Tert*$^{-/-}$ mice.** (A) Illustration of fisetin administration. Starting from 10 weeks of age, mice were given standard mouse feed supplemented with 500 mg/kg of fisetin. Additionally, intermittent gavage administration was performed three days per week at a dose of 20 mg/kg for a total duration of seven weeks. (B–C) Blood biochemistry analysis of alanine transaminase and urea levels in mice from different treatment groups. These parameters reflect liver and kidney function, respectively. (D–E) RT-PCR analysis of *p16*$^{INK4a}$ and *p21*$^{CIP1}$ mRNA expression in kidney tissue of mice from different treatment groups. (F–G) RT-PCR analysis of *p16*$^{INK4a}$ and *p21*$^{CIP1}$ mRNA expression in liver tissue of mice from different treatment groups. (H) Masson staining to evaluate collagen fiber deposition in the kidney tissue of mice from different treatment groups. Scale bar, 50 μm $n = 3$. Data represents the mean ± SEM using unpaired $t$-test.

stress, potentially culminating in inflammation and cell death, underscoring the intricate connection between augmented ROS-induced oxidative stress and the aging process, this indicates a close relationship between increased ROS-induced oxidative stress and aging.

Within the cohort of differentially expressed genes, our research honed in on stanniocalcin 1 (Stc1) due to its pronounced disparities. Stc1 is known to be highly expressed in various tumor tissues and is closely associated with cancer cell proliferation, invasion, apoptosis, and vasculogenesis (*Tang et al., 2014*; *Zhao et al., 2020*). Importantly, prior research has shed light on Stc1's role in mitigating oxidative stress, thereby aiding cells in their resilience against oxidative damage (*Tang et al., 2014*).

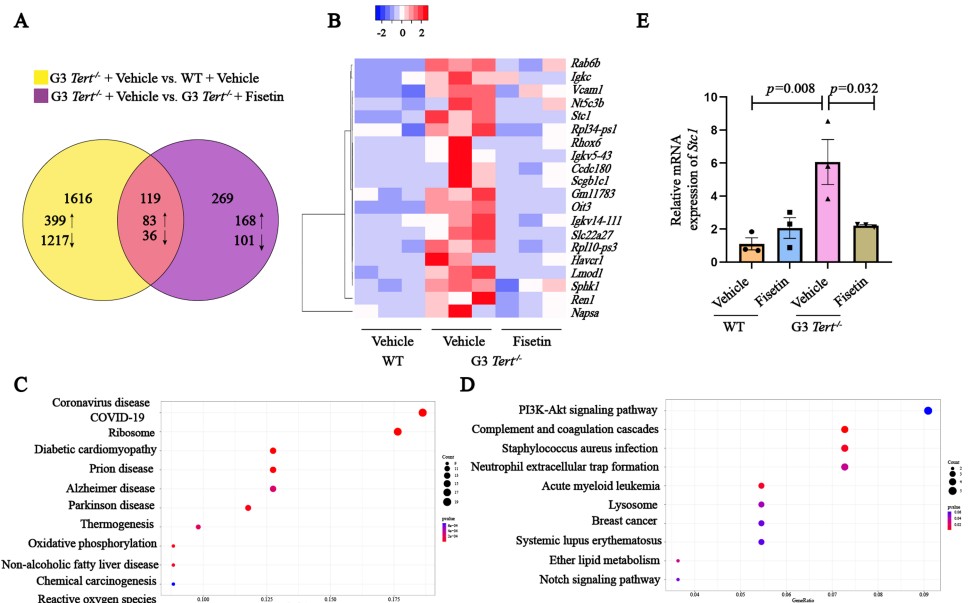

**Figure 3** **Screening of potential targets for anti-aging mechanism of fisetin.** (A) Venn diagram illustrating the overlapping differentially expressed genes between the G3 $Tert^{-/-}$ + Vehicle group and the WT+Vehicle group, as well as between the G3 $Tert^{-/-}$ + Vehicle group and the G3 $Tert^{-/-}$ + Fisetin group. There are a total of 119 genes in this intersection (Q ≤ 0.05, |log2FC| ≥ 0.32). (B) Heatmap displaying the top 20 genes that show high expression in G3 $Tert^{-/-}$ mice and a significant reduction after fisetin treatment. (C) KEGG analysis showing pathways enriched by the upregulated differential genes in the G3 $Tert^{-/-}$ + Vehicle group compared to the G3 $Tert^{-/-}$ + Fisetin group. (D) KEGG analysis showing pathways enriched by the down-regulated differential genes in the G3 $Tert^{-/-}$ + Vehicle group compared to the G3 $Tert^{-/-}$ + Fisetin group. (E) RT-PCR analysis to measure the expression of *Stc1* in kidney tissue of mice from different treatment groups (*n* = 3). Data represents the mean ± SEM using unpaired *t*-test.

Our efforts culminated in an RT-qPCR analysis that unequivocally confirmed the upregulation of *Stc1* expression by approximately 6-fold in G3 $Tert^{-/-}$ mice when compared to the WT group. Significantly, this upregulation was effectively reversed, returning to the original levels following fisetin treatment (Fig. 3E). These compelling findings strongly suggest that fisetin treatment has the potential to exert its anti-aging effects and ameliorate senescence phenotypes by suppressing the expression of Stc1.

## Stc1 plays an anti-apoptotic role by activating Akt signaling pathway in $Tert^{-/-}$ mice

Previous studies have highlighted the role of Stc1 in promoting tumor cell metastasis through the activation of the PI3K/Akt signaling pathway (*Murai et al., 2014*). At the same time, activated Akt can counteract oxidative stress, inhibit cell apoptosis, and protect cells from oxidative damage (*Sha et al., 2021*). Building upon this knowledge, we hypothesized that Stc1 might play a role in promoting the survival of senescent cells by regulating the PI3K/Akt signaling pathway. Western blot analysis revealed that the level of pAkt in G3 $Tert^{-/-}$ mice was approximately 1.3 times higher compared to the WT group. However, after fisetin treatment, the pAkt level was reduced to the baseline level (Figs. 4A and

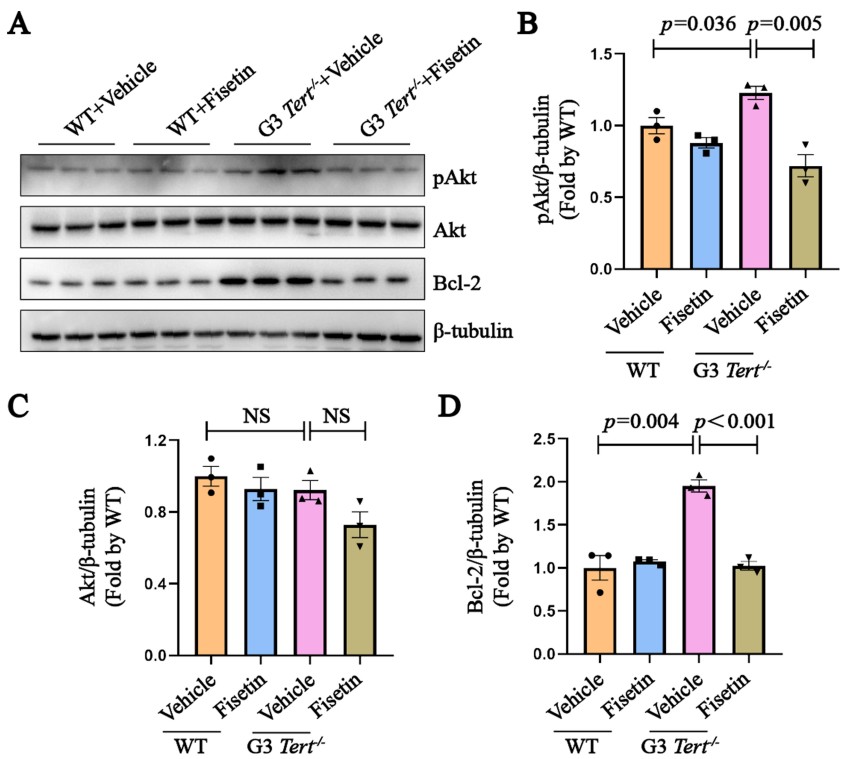

**Figure 4   Stc1 plays an anti-apoptotic role by activating Akt signaling pathway in *Tert*⁻/⁻ mice.** (A) Western blot analysis was performed to evaluate the protein expression levels of Akt, pAkt and Bcl-2 in kidney tissues from mice in different treatment groups. $\beta$-tubulin was served as loading control. (B–D) The protein levels of pAkt, Akt and Bcl-2 were quantified. $n = 3$. Data represents the mean ± SEM using unpaired $t$-test.

4B). There was no change in the total Akt protein level (Fig. 4C). Similarly, Bcl-2, an anti-apoptotic protein downstream of the PI3K/Akt signaling pathway, exhibited similar changes in protein expression as pAkt (Fig. 4D). These results suggest that fisetin may exert its senolytic effects by inhibiting the PI3K/Akt signaling pathway through the suppression of *Stc1*, ultimately improve senescence phenotypes.

## DISCUSSION

Our study provided confirmation of aging characteristics in G3 *Tert*⁻/⁻ mice, including telomere shortening and increased expression of senescence markers in organ tissues (Fig. 1). G1 *Tert*⁻/⁻ mice, on the other hand, did not exhibit significant telomere shortening, possibly due to the presence of telomeres with chromosome end protection functions (*Blasco et al., 1997*). However, due to the fact that we only measured telomere length in one mouse each from the WT, G1 *Tert*⁻/⁻, and G3 *Tert*⁻/⁻ groups, there are limitations in terms of sample size. Mice with telomerase deficiency serve as a robust model for studying anti-aging effects as they closely resemble natural aging and reflect various pathophysiological states during the aging process, making them more representative

compared to other aging mouse models induced by different methods (*Jaskelioff et al., 2011*).

Cellular senescence, a hallmark of aging, refers to the persistent and irreversible cell cycle arrest that proliferating cells undergo in response to endogenous stimuli or environmental pressures (*Di Micco et al., 2021*). Although there is no universally accepted specific marker to identify senescent cells, they can be identified by a combination of common features, including positive staining for SA-$\beta$-gal (*Debacq-Chainiaux et al., 2009*), upregulation of CDK inhibitors (particularly $p16^{INK4a}$ and $p21^{CIP1}$), DNA damage accumulation, formation of senescence-associated heterochromatin foci, elevated intracellular ROS levels, and high levels of SASP factors (*Coppe et al., 2008*), among others (*Lopez-Otin et al., 2023*; *Summer et al., 2019*). Targeted clearance of senescent cells has been shown to extend lifespan and represents an important strategy for intervening in aging and age-related diseases.

Senolytic drugs have demonstrated significant efficacy in improving cardiovascular, renal, hepatic, and pulmonary functions (*Kirkland & Tchkonia, 2020*). There is a growing need for the development of highly specific, safe, and minimally toxic senolytic drugs with great potential for clinical translation. Fisetin, as a naturally occurring compound derived from plants, currently has no reported adverse effects and holds promise as a senolytic drug candidate. Our study confirmed the significant inhibition of aging-related gene upregulation and collagen fiber deposition in kidney tissues of G3 $Tert^{-/-}$ mice with fisetin treatment (Fig. 2). Furthermore, fisetin effectively reduced urea and alanine transaminase levels (Fig. 2), indicating its protective effect on age-related kidney and liver functions. These results collectively demonstrate the effective anti-aging properties of fisetin.

RNA-Seq analysis unveiled a total of 83 upregulated genes in the G3 $Tert^{-/-}$ mice, and their expression profiles exhibited a notable reversal post fisetin treatment. Notably, a heatmap spotlighted the top 20 genes demonstrating the most profound changes. A comprehensive KEGG analysis, comparing the differentially expressed genes in the G3 $Tert^{-/-}$ group *versus* the G3 $Tert^{-/-}$ + Fisetin group, illuminated critical pathways significantly enriched in oxidative phosphorylation, reactive oxygen species dynamics, and complement and coagulation cascades (Fig. 3, Fig. S2).

Among this cohort of genes, stanniocalcin 1 (Stc1) emerged as a compelling focus of our investigation. Stc1, recognized as a glycoprotein hormone, commands heightened expression in various tumor tissues and is intricately associated with pivotal cellular processes, encompassing cell proliferation, invasion, oxidative stress modulation, and apoptosis (*Lin et al., 2022*; *Pena et al., 2013*; *Xiong & Wang, 2019*).

Furthermore, our exploration delved into the PI3K/Akt pathway, an intricate signaling cascade renowned for its multifaceted roles in oxidative phosphorylation, cellular metabolism, proliferation, and cell survival. This pathway has captivated the medical research community for its paramount significance. Evidence from prior studies has underscored Stc1's role in orchestrating the invasion and metastasis of gastric cells through modulation of the PI3K/Akt signaling pathway (*Wang et al., 2019*). Within the context of our study, we observed a substantial elevation in pAkt protein levels within kidney tissues of G3 $Tert^{-/-}$ mice, which intriguingly reverted to levels akin to the control group post

fisetin administration. Moreover, a congruent pattern emerged in the expression of Bcl-2, an anti-apoptotic protein and downstream target of the PI3K/Akt pathway (Fig. 4).

We propose a working hypothesis that fisetin, through the inhibition of Stc1, may orchestrate the attenuation of aging in G3 $Tert^{-/-}$ mice, this could occur *via* subsequent suppression of the PI3K/Akt signaling pathway, culminating in a reduction of excessive ROS production and alleviation of oxidative stress.

## CONCLUSIONS

In this study, we successfully established a *Tert* knockout premature aging mouse model to mimic the natural aging process and investigate potential anti-aging treatments. Our findings demonstrated that fisetin treatment effectively suppressed the upregulation of aging-related genes in the kidneys and livers of G3 $Tert^{-/-}$ mice, while also inhibiting collagen fiber deposition. Through RNA-Seq analysis, we identified *Stc1* as a key factor closely associated with cell survival. Fisetin exerted its senolytic effects by inhibiting the expression of *Stc1*, thereby suppressing the Akt signaling pathway and downstream anti-apoptotic proteins. This mechanism provides a novel insight into the clearance of senescent cells and holds potential for interventions in the aging process and age-related diseases.

### Funding

This work was supported by the National Natural Science Foundation of China (No. 32170762) and the Tianjin Health Research Project (No. 19YFZCSY00600). The funders had no role in study design, data collection and analysis, decision to publish, or preparation of the manuscript.

### Grant Disclosures

The following grant information was disclosed by the authors:
National Natural Science Foundation of China: 32170762.
Tianjin Health Research Project: 19YFZCSY00600.

### Competing Interests

The authors declare there are no competing interests.

### Author Contributions

- Rui Zhao conceived and designed the experiments, performed the experiments, analyzed the data, prepared figures and/or tables, authored or reviewed drafts of the article, and approved the final draft.
- Haomeng Kou conceived and designed the experiments, performed the experiments, analyzed the data, prepared figures and/or tables, and approved the final draft.
- Duo Jiang performed the experiments, authored or reviewed drafts of the article, and approved the final draft.

- Feng Wang conceived and designed the experiments, authored or reviewed drafts of the article, and approved the final draft.

## Animal Ethics

The following information was supplied relating to ethical approvals (i.e., approving body and any reference numbers):

Animal Care and Use Committee of Tianjin Medical University (TMUaMEC2022015).

## DNA Deposition

The following information was supplied regarding the deposition of DNA sequences:

The RNA-Seq sequences are available at NCBI SRA BioSample accession numbers SAMN36759034 to SAMN36759042; PRJNA1000093.

## Data Availability

The original ct value of mRNA expression detected by Real-time PCR and the original images of western blot are available in the Supplemental Files.

## Supplemental Information

Supplemental information for this article can be found online at http://dx.doi.org/10.7717/peerj.16463#supplemental-information.

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
