# Peer review of "Exploring the anti-aging effects of fisetin in telomerase-deficient progeria mouse model"

_PeerJ, doi:10.7717/peerj.16463_

## Round 0.1 · original submission · Major Revisions

Dear Authors,

Both reviewers are of the opinion that your manuscript is suitable for publication in PeerJ. I agree with them that the manuscript is well presented with good figures and tables, and that the writing is clear. We have all found their mouse model very interesting. However there are a number of issues that both reviewers have pointed out, some are minor and some are major, but I think they are all important and need to be resolved before your work can be accepted. Please address all comments from both reviewers and send me a letter with a point-by-point response to all comments as well as a new version of the manuscript with the changes marked or commented.

Reviewer 1 ·

Basic reporting

The paper is generally clear with some ambiguity in the wording. Specifically, the use of baseline can be confusing. Here the authors use "baseline" referring to the WT gene expression level, but audiences may confuse that the baseline refer to the expression level in same subject before fisetin treatment. I suggest the authors change the word to make it more clear.

In terms of citations, authors should add more references when refer to certain knowledge, for example in line 217,phenotype resembling premature aging (add citation) and 225 fisetin as a senolytic drug (add citation).

Experimental design

The research falls within the aim and scope of PeerJ and the experiments are generally well designed. Few minor things I'd like to point out are:
1) in line 215, you stated a significant reduction in telomere length in G3 tert -/- mice. You should use appropriate statistical method to support "significant reduction".
2) I'm curious whether the G3 tert -/- mice exhibit any other observable phenotype of premature aging?
3) You looked at the kidney and the liver. May you elaborate on your choices of the organs to look at?

Validity of the findings

In terms of data transparency, I hope the author can upload their RNA-seq results to GEO database or provide a full list of differentially expressed genes after DEseq2 processing and their filtering criteria.

Additional comments

Overall the paper is of good qualities with minor revisions to be made.

Reviewer 2 ·

Basic reporting

The article is clearly written. The authors are showing a interesting topic with sufficient background. Figures and tables are very well presented.

Experimental design

The article suits the scope of the journal. It does provide a interesting mouse model for future study, but more need to be addressed (see below).
In the material and method, it may seem to be common sense like RIPA buffer or TBST (line 185-193). The recipes do vary from lab to lab, please indicate the recipes you use for better clarity. Line 172-181, if the authors want to adapt protocol from literatures, please indicate the source. the current section is not repeatable with no concentrations stated or instructions to follow.

Validity of the findings

The authors highlighted below as their main findings:
1. We utilized telomerase deficient mouse model to simulate natural aging and validate the anti-
aging effects of fisetin.
Based on the experiment design, the animals used in the manuscript were about 7 month-old. I suspect the difference between WT and Tert-/- mouse in older age (10/12 month) will remain obvious. if not, the model does not provide a very meaning tool. How stable is the genotype? What is the lifespan of this genotype?

2. There were 83 genes that highly expressed in G3 Tert-/- mice, and their expression levels were
restored to baseline after fisetin treatment.
The authors should spend more time digging into the data. It will be more helpful if they can categorize these genes by pathways.

3. Fisetin treatment exerted its anti-aging effects by inhibiting the expression of Stc1, thereby
suppressing the activation of the PI3K/Akt signaling
As mentioned above, the authors need to provide more information about the model before one can really assess the results. Also, if they spend more time in data analysis, it should give a better explanation of how Stc1 responds to Fisetin,

---

## Round 0.2 · Minor Revisions

In general, you have addressed the reviewers' comments and suggestions adequately and your study is almost ready to be accepted for publication. However, please consider reviewer #1's suggestion that you state in the text that there is a limitation due to the small sample size. Once you send it to me it will be ready for acceptance. Thank you

Reviewer 1 ·

Basic reporting

The authors responded well to my previous comment on the use of word and proper references. Now the article is clear with proper reporting.

Experimental design

The authors have responded well to my previous comments and I have no additional revision suggestions.

Validity of the findings

Line 223-225 regarding my previous comment: Since the author only measured one single mouse for telomere length in each group, though the authors referenced to previous literature with proper citation, I still suggest the author note such limitation of small sample size in the text.

Additional comments

The authors provided a clearly formatted and carefully revised version of their manuscript. In general, my previous comments have been addressed. I'm in favor of accepting it after the one minor revision in the text as mentioned above.

Reviewer 2 ·

Basic reporting

The manuscript is well written and well organized.

Experimental design

After the revision, the manuscript is clear on the experimental design.

Validity of the findings

After the revision, the findings are more in depth and logical.

---

## Round 0.3 · accepted · Accept

Thank you very much for addressing adequately all concerns raised by the reviewers. After my assessment, I believe the current version is now ready for publication. Congratulations